# Enhancing Chemotherapeutic Efficacy in Lung Cancer Cells Through Synergistic Targeting of the PI3K/AKT Pathway with Small Molecule Inhibitors

**DOI:** 10.3390/ijms26178378

**Published:** 2025-08-28

**Authors:** Maria Michael, Maria Christou, Iason Kanakas, Christiana M. Neophytou

**Affiliations:** 1Apoptosis and Cancer Chemoresistance Laboratory, Basic and Translational Cancer Research Center, Department of Life Sciences, European University of Cyprus, Nicosia 2404, Cyprus; mm221157@students.euc.ac.cy (M.M.); ik231342@students.euc.ac.cy (I.K.); 2Tumor Microenvironment, Metastasis and Experimental Therapeutics Laboratory, Basic and Translational Cancer Research Center, Department of Life Sciences, European University Cyprus, Nicosia 2404, Cyprus; mc231061@students.euc.ac.cy

**Keywords:** lung cancer, chemotherapy, small molecule inhibitors, PI3K/AKT pathway, combination therapy, Cisplatin, 5-fluorouracil, MK2206, BKM120

## Abstract

Non-small cell lung cancer (NSCLC) remains one of the leading causes of cancer-related mortality, with resistance to chemotherapy representing a major therapeutic challenge. In this study, we investigated the effects of conventional chemotherapeutics, Cisplatin and 5-fluorouracil (5-FU), in combination with small molecule inhibitors (SMIs) targeting the PI3K/AKT signaling pathway, on NSCLC cell viability. Two NSCLC cell lines, H460 (large cell lung carcinoma) and A549 (adenocarcinoma), both characterized by constitutive activation of PI3K/AKT signaling, were evaluated. A normal human lung fibroblast cell line, MRC-5, was used as a non-cancer control to assess selectivity and exclude cytotoxic effects. Dose–response analyses were performed to determine the optimal concentrations of Cisplatin, 5-FU, the AKT inhibitor MK2206, and the PI3K inhibitor BKM120, both as monotherapies and in combination treatments. We identified a synergistic combination of 5-FU and BKM120 that significantly reduced viability and induced apoptosis in NSCLC cells while sparing MRC-5 cells. Mechanistic studies revealed that apoptosis induction was mediated through the apoptotic pathway regulated by the Bcl-2 family and activation of caspase-3 and caspase-6. These findings highlight the therapeutic potential of combining PI3K/AKT inhibitors with conventional chemotherapy to overcome resistance mechanisms in NSCLC.

## 1. Introduction

Lung cancer (LC) is the leading cause of cancer-related deaths in both sexes worldwide. Specifically, LC represents 9.4% of all cancer cases in women and 15.2% in men, with corresponding mortality rates of 13.5% and 22.7% [1]. There are two main types of lung cancer: small cell lung cancer (SCLC) and non-small cell lung cancer (NSCLC). NSCLC is the most common form of lung cancer and grows at a slower rate compared to SCLC. NSCLC has various subtypes, such as adenocarcinoma, squamous cell carcinoma, and large cell carcinoma (LCLC) [2].

There has been notable advancement in the field of lung cancer treatment over the years. The most common treatments in LC patients are surgery, radiotherapy, chemotherapy, immunotherapy, and targeted therapy [3]. Chemotherapy, such as platinum agents (Cisplatin or carboplatin and/or etoposide), is mainly used either as monotherapy or combinational therapy with various drugs. Surgery sometimes followed by chemotherapy (adjuvant) or chemotherapy followed by surgery (neoadjuvant) is the main treatment plan for LC patients [4]. Even though there have been great advances in therapeutic approaches against lung cancer, all the traditional chemotherapeutic drugs have major side effects. Importantly, the development of resistance is still a major obstacle in therapeutic efficacy [5].

Cisplatin, a platinum-based chemotherapy drug, belongs to the class of alkylating agents that have the ability to add alkyl groups to DNA [6,7]. As a result, the repair enzymes, in their attempt to replace the alkylated bases, fragment the DNA, thereby preventing both DNA synthesis and RNA transcription. Resistance to Cisplatin may arise because of p-glycoprotein-mediated drug efflux, detoxifying systems, and DNA repair mechanisms [8]; in addition, the PI3K/Akt/mTOR axis functions as a key modulator of drug resistance [9]. 5-Fluorouracil (5-FU) is a fluoronucleotide, a structure that resembles a pyrimidine–pyrimidine analog. The misincorporation of fluoronucleotides into RNA and DNA, as well as the suppression of the enzyme thymidylate synthase, has been linked to the cellular toxicity of 5-FU [10]. Even though it is not standard in lung cancer treatment protocols, it is used experimentally or in combination therapies, particularly in clinical trials or second/third-line settings [11,12]. Resistance to 5-FU has been reported in many cancer types, including lung [13]. The PI3K/AKT pathway has been identified to be a major contributor of 5-FU resistance in lung cancer; the potential therapeutic targeting of this pathway in an effort to reverse resistance has been widely reported [14,15].

Studies have shown that BKM120, an oral, highly selective pan-class I PI3K inhibitor, when combined with other drugs like rapamycin, gefitinib, and everolimus, provides synergistic anti-tumor effects, especially in lung cancer mouse models and patients with mutations in the PI3K pathway [16,17,18,19]. MK2206 is a selective allosteric inhibitor of AKT, a key serine/threonine kinase involved in several cellular processes, including cell survival, growth, and metabolism [20]. Moreover, pre-clinical studies suggested that MK2206, in combination with other targeted therapies like mTOR inhibitors or Epidermal Growth Factor Receptor (EGFR) inhibitors, can enhance anti-tumor activity and overcome resistance to single-agent treatments [21,22].

The purpose of the present study was to evaluate whether a combination of Cisplatin or 5-FU with small molecule inhibitors (SMIs) of the PI3K/AKT pathway, BKM120 and MK2206, would re-sensitize lung cancer cells to chemotherapy and improve its efficacy. We used two NSCLC cell lines of different origin, H460 and A549, as well as human lung fibroblast cells MRC-5 to exclude toxic effects. We determined the optimal combination of agents and their concentrations that show synergistic action in lung cancer cells without affecting normal cells. Their mechanism of action did not involve the production of reactive oxygen species (ROS) but rather the inhibition of the AKT pathway and the activation of caspases and induction of apoptosis. Improving treatment regimens may lead to increased efficacy of drugs at lower dosages, thereby enhancing efficacy and lowering negative side effects.

## 2. Results

### 2.1. Combination of Cisplatin or 5-FU with Small Molecule Inhibitors Improves Their Cytotoxicity in Lung Cancer Cells

To investigate the anti-proliferative effect of chemotherapy and SMIs in lung cancer cell lines, we performed the MTT viability assay. H460 lung cancer cells were treated with Cisplatin, 5-FU, MK2206, and BKM120 as monotherapies in a dose- and time-dependent manner. Monotherapy with Cisplatin (1–30 μM) reduced cell viability significantly at 48 and 72 h compared to 24 h, with no additional decrease after 48 h (Figure 1A). Similarly, 5-FU (10–100 μM) reduced the cell viability in a time-dependent manner, with a 20% reduction at 24 and 48 h and an additional 30% decrease at 72 h (Figure 1B). MK2206 (0.1–5 μM) showed the greatest cell viability reduction at 24 h, but a restoration in cell viability was observed at 72 h, suggesting an inactivation of the drug in prolonged incubation times (Figure 1C). BKM120 (0.1–30 μΜ) reduced cell viability at 24 and 48 h almost across all concentrations and time points (Figure 1D).

The effects of Cisplatin, 5-FU, MK2206, and BKM120 individually were also evaluated in A549 cells to assess their impact on cell viability. Treatment with Cisplatin (1–30 μM) demonstrated minimal effects at 24 h but showed a concentration-dependent decrease in viability at 48 and 72 h, with 30 μM being the most effective concentration (Figure 2A). Similarly, treatment with 5-FU (10–100 μM) showed progressive reduction in viability across all the time points, with the greatest decline observed at 72 h (Figure 2B). MK2206 (0.1–5 μM) induced a dose-dependent reduction in viability at 24 and 48 h (Figure 2C). BKM120 (0.1–30 μM) reduced viability at all concentrations except 0.5 μM BKM120 (Figure 2D).

The IC50 indicates the drug concentration required to achieve 50% inhibition of a biological process. Table 1 presents the IC50 values for Cisplatin, 5-FU, MK2206, and BKM120 in H460 and A549, respectively, after 24, 48, and 72 h of incubation. The data reveal that the IC50 for all agents generally decreases with longer incubation times, suggesting increased efficacy. However, MK2206 deviates from this trend, as its IC50 increases from 117.5 μM at 24 h to 202.3 μM at 48 h.

### 2.2. Chou–Talalay Analysis Reveals Synergistic Interactions Between Drug Concentrations

Subsequently, combination treatments were evaluated with the MTT assay, and their Combination Index (CI) was calculated using the Chou–Talalay method. In H460 cells, the combination of 10 μM Cisplatin with 1 μΜ ΒΚΜ120 at 48 h was the most effective in producing synergistic effects (Figure 3A, Table 2). Moreover, the combination of 10 μM 5-FU with BKM120 (0.1–1 μM) reduced the viability at 48 h more effectively than monotherapy (Figure 3B, Table 2). Based on our viability data (Figure 1B,C), we also chose to combine 10 μM 5-FU with 1 μM MK2206; these concentrations were selected as they were effective in monotherapy without reducing viability below 50%. The combination of those two drugs showed synergism in reducing the viability of H460 cells at both 24 and 48 h (Figure 3C). Additional combinations that did not show a synergistic effect can be found in Appendix A.

Certain combination treatments in A549 cells were more effective than monotherapies. Concentrations were selected based on the viability graphs (Figure 2) so that viability is reduced no more than 50%. The combined treatment of 10 μΜ 5-FU with either 1 μΜ MK2206 (Figure 4A) or BKM120 (Figure 4B) displayed enhanced effects and improved the anti-cancer activity of the chemotherapy. Cisplatin (10 μM) combined with either 1 μΜ MK2206 (Figure 4C) or 1 μM BKM120 (Figure 4D) decreased viability by approximately 50%.

We further characterized these pharmacological interactions using the Chou–Talalay method and calculated the Combination Index (CI). Synergism—the capacity of the drugs to work together more successfully than would be expected—is indicated by a CI of less than 1. An antagonistic (anti-drug) interaction between the medications is indicated by a CI larger than 1. Additionally, when 0.9 < CI < 1.1, the overall impact is additive.

The most effective combinations are shown below in Table 2 and Table 3. Additional CI values can be found in Appendix A. In H460 cells, the combination of Cisplatin (10 μM) with varying concentrations of BKM120 (0.1 μM, 0.5 μM) yielded CI values greater than 1, indicating an antagonistic effect; however, the combination of 10 μM Cisplatin and 1 μM BKM120 showed CI = 0.22, indicating strong synergistic action. Additionally, CI values were determined for the combination of 10 μM 5-FU with BKM120 (0.1 μM, 0.5 μM, and 1 μM). Although the combinations after 24-hour incubation shows antagonistic effect (CI > 1) (Appendix A), the compounds following 48 h incubation showed synergistic effect (CI < 1) in all combinations, with 10 μM 5-FU in combination with 0.1 μM BKM120, displaying the best synergistic effect (Table 2).

A similar approach was followed in A549 cells, and the CI values were calculated (Table 3). The combination of 10 μΜ Cisplatin with either 1 μM BKM120 or MK2206 produced moderate synergistic effects, with a CI of 0.48 and 0.88, respectively. A more enhanced effect was observed when cells were incubated with the same concentrations of 5-FU and SMIs; 10 μΜ of 5-FU with 1 μΜ ΒΚΜ120 synergistically reduced the viability of A549 cells (Figure 4B) with a CI of 0.23, while a similar effect was observed with 1 μΜ MK2206 (Table 3).

### 2.3. Combination of Selected Concentrations of 5-FU and BKM120 in H460 Cells Induces Apoptosis Without Affecting Normal Lung Cells

To further investigate the efficacy and mechanism of action of a potent, selected combination in lung cancer cells, we incubated H460 with 10 μM 5-FU and 0.1 μM BKM120 and performed the Annexin V/PI staining assay for apoptosis. Apoptosis was evaluated by flow cytometry (Figure 5A). Each panel represents a different condition: in the lower left, viable cells (unstained); lower right, early apoptotic cells (Annexin V+); upper right, late apoptotic cells (Annexin V+/Propidium Iodide+); and upper left, necrotic cells (Propidium Iodide+) are shown, respectively. As can be seen from the analysis (Figure 5A), the incubation with 0.1 μΜ BKM120 reduces cell viability from 94% to 78.4%. There is a significant decrease in the percentage of viable cells in cells treated with 5-FU compared to the control group, down to 26.5%. In addition, the percentage of Annexin V+ cells, including early and late apoptotic cells, increases compared to the control from 4.5% to 13.3% in BKM120-treated cells and to 6.3% following incubation with 5-FU. The combination therapy produced the best results compared to the control and the monotherapies. The percentage of viable cells in combination therapy was 11.6%, and of apoptotic cells, 13.5%. The percentage of necrotic cells in the combination treatment is 74%, which is significantly higher than both monotherapies; however, this may indicate a non-apoptotic or necrotic pathway of cell death that warrants further investigation.

Importantly, the combination of 10 μM 5-FU and 0.1 μM BKM120 that showed effectiveness in lung cancer cells did not affect the viability of MRC-5 normal human lung fibroblast cells (Figure 5B). 5-FU induced a cytotoxic effect in MRC-5, reducing the viability to approximately 40%. Co-incubation of cells with BKM120 reversed this effect, suggesting a protective mechanism that may be potentially explored in therapeutic regimens. To further exclude any non-specific effects of the combination therapy, we measured the ability of 5-FU and BKM120 to produce ROS using the DCFH-DA assay. We found that the agents alone or in combination do not increase the DCFH signal when added in the H460 cell line (Figure 5C), further supporting a more specific, apoptotic effect.

### 2.4. The Combination of Selected Agents Modifies the Expression Levels of Pro- and Anti-Apoptotic Proteins in H460 Cells

To further investigate the underlying mechanism of action of 5-FU in combination with BKM120 in H460 cells, the expression levels of key proteins implicated in apoptotic pathways were investigated. H460 cells were incubated for 48 h with 10 μM 5-FU, 0.1 μM BKM120, and their combination. Pro-apoptotic Bax mRNA and protein levels were increased more prominently after exposure to the combination treatment (Figure 6A,B). AKT, p-AKT, and Bcl-2 proteins were analyzed by Western blot. The results indicate a reduction in p-AKT protein expression levels in all conditions, with the most pronounced decrease observed in the combination treatment (Figure 6C). AKT expression levels remain stable. Additionally, the protein and mRNA levels of anti-apoptotic protein Bcl-2 were notably decreased in the combination treatment (Figure 6C and Appendix A). The expression levels of Caspases 3 and -6 were decreased in 5-FU and in combination treatment (Figure 6D). Alpha-Fodrin, a cytoskeletal protein processed by caspase-3 during apoptosis, was cleaved under treatment conditions, especially in the presence of both drugs (Figure 6D).

## 3. Discussion

In this study, H460 and A549 lung cancer cell lines were used to investigate the effects of chemotherapy, Cisplatin, and 5-FU, along with small molecule inhibitors (SMIs) of PI3K/AKT in cell viability. MRC-5, a normal human lung fibroblast cell line, was used as a control to exclude cytotoxicity. H460 is classified as large cell lung carcinoma, while A549 cells are derived from human lung adenocarcinoma; both are subtypes of NSCLC. Both cell lines exhibit constitutive activation of the PI3K/AKT signaling pathway, which is a major driver of cell survival, proliferation, and epithelial-to-mesenchymal transition (EMT). Since chemotherapy primarily exerts its effects by inducing DNA damage, and cancer cells often evade this damage through activation of survival pathways such as PI3K/AKT ([23,24]), we hypothesized that concurrent inhibition of PI3K/AKT would suppress pro-survival signaling and thereby enhance chemotherapy-induced apoptosis. In addition, both compounds, MK2206 and BKM120, have been reported to lower chemoresistance, acting synergistically with other agents in pre-clinical studies [25,26,27,28]. We determined the optimal concentration of Cisplatin, 5-FU, AKT inhibitor MK2206, and PI3K inhibitor BKM120, alone and in combination treatments. The optimal combination (5-FU and BKM120) selectively induced apoptosis in lung cancer cells without affecting normal cells. The mechanism of action involved the Bcl-2 family-controlled apoptotic pathway and activation of caspase-3 and caspase-6.

Initially we investigated the anti-proliferative effects of Cisplatin as monotherapy in H460 (Figure 1A) and A549 (Figure 2A). The mechanism of action of Cisplatin’s cytotoxicity involves the addition of alkyl groups to DNA, the induction of double-stranded DNA breaks, and the promotion of apoptosis [29]. Although Cisplatin therapy is one of the most effective agents used to treat various types of cancer, it has been associated with many side effects, such as hepatotoxicity and nephrotoxicity [30,31]. 5-FU is frequently used in cancer treatment, either alone or in combination with other treatment plans [32]. In spite of innovations over the last two decades, resistance is still the major drawback for 5-FU clinical application [13]. Therefore, combination treatments concurrently targeting aberrantly activated pathways may increase efficacy, reduce effective concentration, and minimize side effects. Cisplatin IC50 at 48 h was calculated at 8.6 μΜ and ~37 μΜ, respectively, similar to other studies (Table 1) [33]. 5-FU IC50 values were found to be 105.2 and 110.2 μΜ in H460 and A549, respectively, at 48 h; the reported IC50 values for both cell lines vary greatly in the literature [34,35].

We also studied the monotherapy effects of BKM120 and MK2206 in both cell lines. MK2206 inhibits the activation of AKT, which is commonly hyperactivated in various types of cancer, contributing to uncontrolled cell proliferation and survival [36]. MK2206 induced a moderate decrease in the viability of both cell lines (Figure 1C and Figure 2C). Interestingly, at 72 h, its effects seem to be reduced, suggesting an inactivation of the drug during prolonged incubation times. Even though this has not been reported in the literature, it may be an underlying cause of the moderate effects the inhibitor has shown in clinical trials [37,38]. MK2206 promotes apoptosis through PI3K/AKT/mTOR inhibition; however, a study reported that AKT1 inhibition by MK2206 led to increased invasion and metastasis of NSCLC cells with K-RAS or EGFR mutations [39,40]. Therefore, its cancer-promoting properties in subsets of lung cancer should be further investigated. The best combination effect using MK2206 (1 μΜ) in our study was observed in A549 cells with 10 μΜ 5-FU (Table 3). MK2206 has been previously reported to exert a synergistic cytotoxic effect against breast cancer cells regardless of their estrogen receptor (ER) and HER2neu status [41]. In addition, the combination of MK2206 with the anti-EGFR monoclonal antibody cetuximab in cetuximab-resistant NSCLC cells decreased the activity of both AKT and MAPK, thus highlighting the importance of simultaneous pathway inhibition [21].

BKM120 is a pan-class I PI3K inhibitor that targets the p11α/β/δ/γ catalytic subunits that has been investigated in clinical trials against advanced cancers [42,43]. Following the completion of pre-clinical and clinical trials, BKM120 demonstrated strong inhibitory action on AKT, leading to cytotoxic and anti-proliferative effects in both hematological malignancies and solid tumors [44,45,46,47]. We found that BKM120 effectively reduces the viability of lung cancer cell lines (Figure 1D and Figure 2D). The combination of 0.1 μΜ ΒΚΜ120 with 10 μΜ 5-FU was selected for further investigation since it showed a strong synergistic effect in H460 cells (CI = 0.08) (Table 2).

The drug combination induced apoptosis in H460 cells (Figure 5A). BKM120 induces pro-apoptotic PUMA activation due to its inhibitory effects on the AKT/Fox03a pathway, crucial for its therapeutic effects [48,49]. Importantly, normal lung cancer cells were not affected by the treatment, while the co-incubation of BKM120 protected cells from 5-FU cytotoxicity (Figure 5B). The potential cytoprotective effect observed when combining small molecule inhibitors (SMIs) targeting PI3K/AKT with 5-FU may be explained by SMI-induced G0/G1 cell cycle arrest, which reduces the proportion of cells undergoing DNA synthesis, thereby diminishing the efficacy of S-phase-specific agents such as 5-FU [50,51]. Normal lung fibroblasts (MRC-5) are predominantly quiescent and maintain tighter regulation of the G1/S checkpoint compared to cancer cells, which often exhibit constitutive PI3K/AKT signaling, defective checkpoint control, and continuous cycling [52]. As a result, BKM120-induced G0/G1 arrest in MRC-5 cells likely restricts their entry into S-phase, thereby limiting the cytotoxic effect of the S-phase-specific agent 5-FU. In contrast, NSCLC cells, which display deregulated proliferation and impaired checkpoint function, remain susceptible to DNA damage even under conditions of PI3K inhibition, leading to apoptosis [53]. MRC-5 fibroblasts exhibit a lower proliferative rate and a higher proportion of cells in the G1 phase compared to H460 lung cancer cells [54,55,56]. This disparity in cell cycle dynamics may contribute to the differential response observed.

Furthermore, cell death did not involve the production of ROS (Figure 5C), which is critical since modulation of ROS may be one mechanism by which cancer cells avoid the cytotoxicity induced by 5-FU [57]. 5-FU primarily acts as an antimetabolite that inhibits thymidylate synthase, thereby disrupting DNA synthesis and inducing DNA damage ([13]). However, tumor cells often evade 5-FU-induced cytotoxicity by activating survival signaling pathways, including PI3K/AKT ([58]). Blocking PI3K by using BKM120 reduced pro-survival signaling (Figure 6C) and lowered anti-apoptotic defenses such as Bcl-2 family upregulation (Figure 6A,B); others have reported that BKM120 impairs DNA damage repair capacity ([59]). By inhibiting PI3K, tumor cells become more vulnerable to DNA damage and stress induced by 5-FU, thereby shifting the balance toward apoptosis. Combination treatment also increased the Bax/Bcl-2 ratio and activated capsase-3 and caspase-6 leading to α-Fodrin cleavage (Figure 6D), which are considered established markers of apoptosis [8]. Synergism between BKM120 and other agents in inducing apoptotic effects has been previously reported [26,55].

## 4. Materials and Methods

### 4.1. Cell Culture and Reagents

H460 and A549 cell lines were obtained from the American Type Culture Collection (ATCC) (Manassas, VA, USA). MRC-5 cells were a kind gift from Dr. Sotiris K. Hadjikakou at the University of Ioannina. H460, A549, and MRC-5 cells were cultured in Roswell Park Memorial Institute (RPMI) in the presence of 10% Fetal Bovine Serum (FBS) and 1% antibiotics/antimycotics. FBS, antibiotic/antimycotic, and trypsin were purchased from Gibco, Invitrogen (Carlsbad, CA, USA). Cisplatin, 5-FU, MK2206, and BKM120 compounds were purchased from Selleck Chemicals (Houston, TX, USA). Antibodies were purchased from Cell Signaling Technology (Danvers, MA, USA). All other reagents were purchased from Sigma Aldrich (St. Louis, MO, USA).

### 4.2. MTT Assay

A total number of 5 × 10^4^ cells were seeded per well of a 96-well plate. The lung cancer cell lines, H460 and A549, were incubated overnight to allow for cell attachment and recovery. For the H460 cell line, cells were treated with varying concentrations of Cisplatin, 5-fluorouracil, MK2206, BKM120, and specific combinations of these agents. Cell viability was assessed using the MTT assay, which measures the reduction of 3-(4,5-dimethylthiazol-2-yl)-2,5-diphenyltetrazolium bromide (MTT) to an insoluble formazan product by metabolically active cells. At the end of each incubation period, 20 μL of MTT dye (1 mg/mL; Sigma, St. Louis, MO, USA) was added in each well, and the plates were incubated at 37 °C for 3 h. The media was then removed, and 200 μL of Dimethyl Sulfoxide (DMSO) were added to dissolve the formazan crystals. Subsequently, the plates were placed on a shaker for 15 min, and absorbance was measured at 570 nm using a microplate reader (VarioSkan, Thermo Fisher Scientific, Waltham, MA, USA). The absorbance values were directly proportional to the number of viable cells per cell. The percentage of cell viability for each treatment group was calculated after normalization to its respective control group. Vehicle control (PBS) was added at the same concentrations as the treatments.

### 4.3. DCFH-DA

2′-7′-Dichloroflurescein diacetate (DCFH-DA) is a lipophilic, non-fluorescent, cell-permeable redox probe. The DCFH-DA readily crosses the cell membrane through passive diffusion followed by deacetylation. The deacylated product is an oxidant-sensitive 2′-7′-Dichloroflurescein (DCFH). DCFH is oxidized later to form highly fluorescent DCF, which is measured at excitation 485 nm/emission 535 nm. To measure inhibition of endogenous ROS, 4 × 10^5^ cells per well were seeded in a 96-well plate and left to attach for 24 h. Cells were treated with 10 μM 5-FU and 0.1 μM BKM120 alone and in combination for 48 h. Following this, 1 μL of DCFH-DA (100 mM) was added to each well, and plates were covered and left to incubate at 37 °C for 1 h. Media without drugs was used as a negative control, while media with DCFH-DA only was used as a positive control. Media was removed, and cells were washed twice with PBS. At the end of the washes, 100 μL of PBS were added in each well, and fluorescence was measured by a plate reader at 485 nm excitation and 535 nm emission for 10 and 30 min.

### 4.4. Annexin V/Propidium Iodide Staining

A total of 1 × 10^5^ H460 cells were seeded per well of a 6-well plate and treated with 10 μM 5-FU, 1 μm BKM120, and their combination and incubated for 48 h. Following the incubation period, cells were harvested and stained according to the protocol provided by the Alexa Fluor™ 488 Annexin V/Dead Cell Apoptosis Kit (Life Technologies, Carlsbad, CA, USA). Cell viability, apoptosis, and cell death were assessed using the Attune NxT Flow Cytometer (Thermo Fisher Scientific, Waltham, MA, USA). Data were analyzed with FlowJo software 10.10.0 (BD Biosciences, Franklin Lakes, NJ, USA), whereas the analysis categorized the cells as follows: Annexin V-positive/Propidium Iodide (PI)-negative cells were classified as early apoptotic cells, Annexin V-positive/PI-positive cells were classified as late apoptotic or dead cells, and Annexin V-negative/PI-negative cells were identified as viable cells.

### 4.5. Total RNA Preparation and Real-Time Quantitative PCR (q-PCR)

Total RNA was isolated using Trizol™ reagent (Invitrogen, Carlsbad, CA, USA) according to the manufacturer’s protocol. Then, complementary DNA (cDNA) was synthetized from the extracted RNA using random primers and the Superscript™ III Reverse Transcriptase kit (Invitrogen, Carlsbad, CA, USA). Primers were designed using Primer3 software version 2.6.1. (accessed on 11 April 2024), and the sequences used for the analysis were as follows: human Bax, 5′-TCAGGATGCGTCCACCAAGAA-3′ (forward), 5′-TGTGTCCACGGCGGCAATCATC-3′ (reverse), and human GAPDH 5′-TTGGTATCGTGGAAGGACTCA-3′ (forward), 5′-TGTCATCATATTTGGCAGGTTT-3′ (reverse). Real-time PCR was performed using the BioRad CFX96 Real-Time System and the SYBR Green PCR Master Mix (Applied Biosystems, Waltham, MA, USA) according to the manufacturer’s instructions. The relative expression levels of target genes were normalized to those from GAPDH amplification using the ΔΔCt method.

### 4.6. Western Blot

Following incubation at indicative time points, cells were washed twice with PBS and lysed in RIPA buffer. The RIPA buffer composition included 150 mM NaCl, 50 mM Tris, 5 mM EDTA [Na2], 1% (*v*/*v*) Triton X-100, 1% (*w*/*v*) deoxycholate (24 mM), and 0.1% (*w*/*v*) SDS (35 mM) supplemented with protease and phosphatase inhibitors (Complete Mini, Roche, Basel, Switzerland) to ensure effective cell membrane disruption. Total protein lysates were collected, and the protein concentration in each sample was determined using the Bradford assay. Proteins were separated via SDS-PAGE electrophoresis and subsequently transferred to PVDF membranes (Merck Millipore, Burlington, MA, USA) for Western blot analysis. Membranes were incubated with SuperSignal West Femto-Substrate (Thermo Scientific, Waltham, MA, USA) per manufacturer’s instructions and visualized using BioRad Universal Hood II and the Image Lab 5.0 software. Densitometry analysis of the Western blot results was performed using ImageJ software v.1.53e. The intensity of protein bands was normalized to the corresponding loading control to ensure accurate quantifications.

### 4.7. Statistical Analysis

Data were presented as the mean ± standard deviation of at least three independent experiments. Statistical analyses were performed using GraphPad Prism v8.4.3 (GraphPad Software). Statistical comparisons between the two groups were conducted using the unpaired *t*-tests. For all tests, a *p*-value < 0.05 was considered statistically significant. The effectiveness of the combinations of drugs was evaluated by the Chou–Talalay method. The general equation of the Chou–Talalay contains the combination of the Michaelis–Menten, Scatchard, Henderson–Hasselbalch, and Hill equations [60].

## 5. Conclusions

In this study, we demonstrated that combined targeting of the PI3K/AKT signaling pathway with conventional chemotherapeutics can effectively enhance cytotoxicity in non-small cell lung cancer (NSCLC) models while sparing normal lung fibroblasts. Using H460 (large cell lung carcinoma) and A549 (adenocarcinoma) cell lines, both characterized by constitutive activation of PI3K/AKT, we systematically evaluated Cisplatin, 5-fluorouracil (5-FU), and small molecule inhibitors (SMIs) of AKT (MK2206) and PI3K (BKM120). We identified a synergistic combination of 5-FU and BKM120 that selectively induced apoptosis in NSCLC cells, mediated by Bcl-2 family-regulated apoptotic signaling and activation of caspase-3 and caspase-6. These findings underscore the potential of integrating PI3K/AKT-targeted therapies with standard chemotherapy to overcome resistance mechanisms in NSCLC.

Future research should focus on the evaluation of detoxifying and adaptive resistance mechanisms that may attenuate the efficacy of SMIs in prolonged treatment regimens, including the role of drug efflux transporters, metabolic enzymes, and intracellular drug sequestration. Additionally, as the A549 and H460 models used in this study harbor wild-type EGFR, it will be critical to investigate whether EGFR mutational status influences sensitivity to PI3K/AKT inhibition. Comparative studies using EGFR-mutant NSCLC cell lines or patient-derived models may provide valuable insights into context-specific therapeutic responses. A lung cancer cell line that has been developed to be resistant to 5-FU should also be investigated to further validate the mechanistic role of the PI3K/AKT pathway in the observed response. Furthermore, our study is an in vitro proof-of-concept to identify synergistic interactions between chemotherapy and PI3K/AKT inhibition in NSCLC. The observed selectivity of the 5-FU and BKM120 combination provides a strong rationale for further work in a more physiologically relevant system, such as NSCLC-derived organoids or animal models.

Emerging strategies, such as combining PI3K/AKT inhibitors with immunotherapies (e.g., immune checkpoint blockade), could further enhance therapeutic efficacy by modulating tumor–immune interactions. Multi-omics profiling—integrating transcriptomic, proteomic, and phospho-proteomic analyses—may identify predictive biomarkers of response and resistance. Finally, validating these drug combinations in in vivo models, including patient-derived xenografts (PDX) and organoid systems, will be an essential step toward clinical translation.

## Figures and Tables

**Figure 1 ijms-26-08378-f001:**
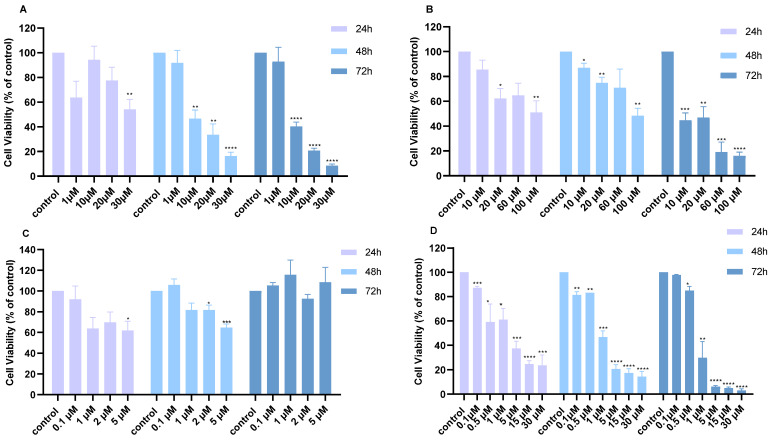
Cell viability of H460 lung cancer cells following treatment with different monotherapies over 24, 48, and 72 h. Cell viability after treatment with increasing concentrations of (**A**) Cisplatin, (**B**) 5-FU, (**C**) MK2206, or (**D**) BKM120. *p*-values: * <0.05, ** <0.01, *** <0.001, **** <0.0001 compared to control. The experiment was carried out in triplicate.

**Figure 2 ijms-26-08378-f002:**
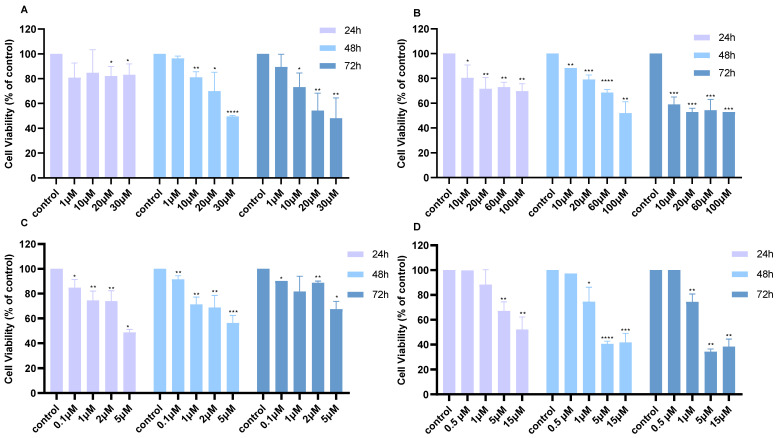
Cell viability of A549 lung cancer cells following treatment with different monotherapies over 24, 48, and 72 h. Cell viability after treatment with (**A**) increasing concentrations of Cisplatin, (**B**) 5-FU, (**C**) MK2206, or (**D**) BKM120. *p*-values: * <0.05, ** <0.01, *** <0.001, **** <0.0001 compared to control. The experiment was carried out in triplicate.

**Figure 3 ijms-26-08378-f003:**
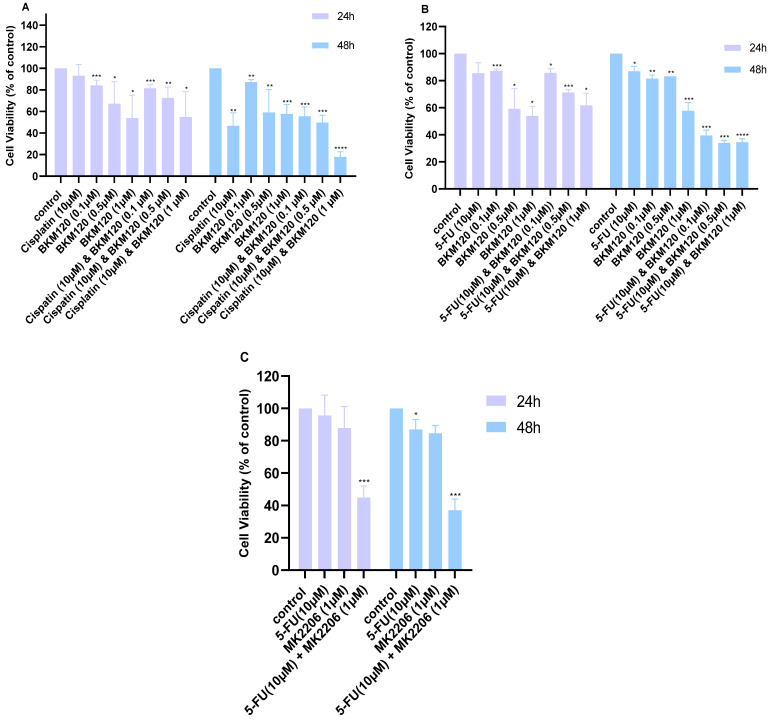
Cell viability of H460 lung cancer cells following treatment with combination therapies. Cell viability was measured following combination treatments at 24 and 48 h. (**A**) Cell viability after administration of 1 and 10 μM Cisplatin with different concentrations of BKM120, (**B**) treatment with 10 μM 5-FU with varying concentrations of BKM120, and (**C**) treatment with 10 μM 5-FU and 1 μM MK2206. *p*-values: * <0.05, ** <0.01, *** <0.001, **** <0.0001 compared to control. The experiment was carried out in triplicate.

**Figure 4 ijms-26-08378-f004:**
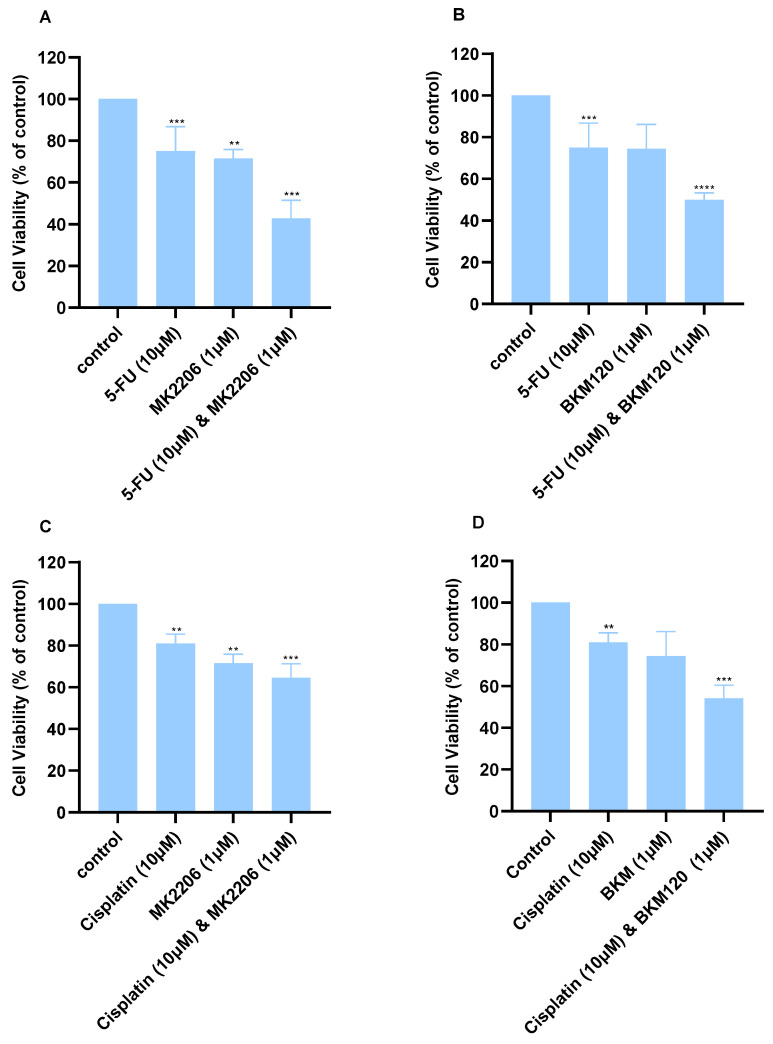
Cell viability of A549 lung cancer cells following treatment with combination therapies. Cell viability following combination treatments for 48 h of (**A**) 10 μM 5-FU and 1 μM MK2206, (**B**) 10 μM 5-FU and 1 μM BKM120, (**C**) 10 μM Cisplatin and 1 μM MK2206, and (**D**) 10 μM Cisplatin and 1 μM BKM120. *p*-values: ** <0.01, *** <0.001, **** <0.0001 compared to control. The experiment was carried out in triplicate.

**Figure 5 ijms-26-08378-f005:**
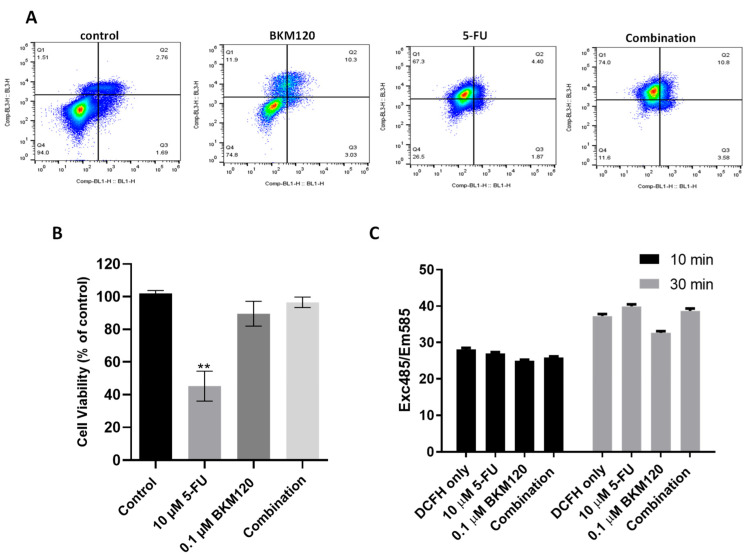
Apoptotic effect of 10 μM 5-FU and 0.1 μM BKM120 selectively in cancer cells. (**A**) H460 cells were treated for 48 h, followed by Annexin V/PI staining and flow cytometry analysis. (**B**) MRC-5 normal lung cells were treated for 48 h, followed by the MTT viability assay. (**C**) Fluorescence signal (485em/585Ex) in the presence of DCFH-DA was measured at 30 and 60 min to detect the presence of ROS following incubation of H460 cells with the agents in monotherapy and combination for 48 h. *p*-values: ** <0.01, compared to control. The experiment was carried out in triplicate.

**Figure 6 ijms-26-08378-f006:**
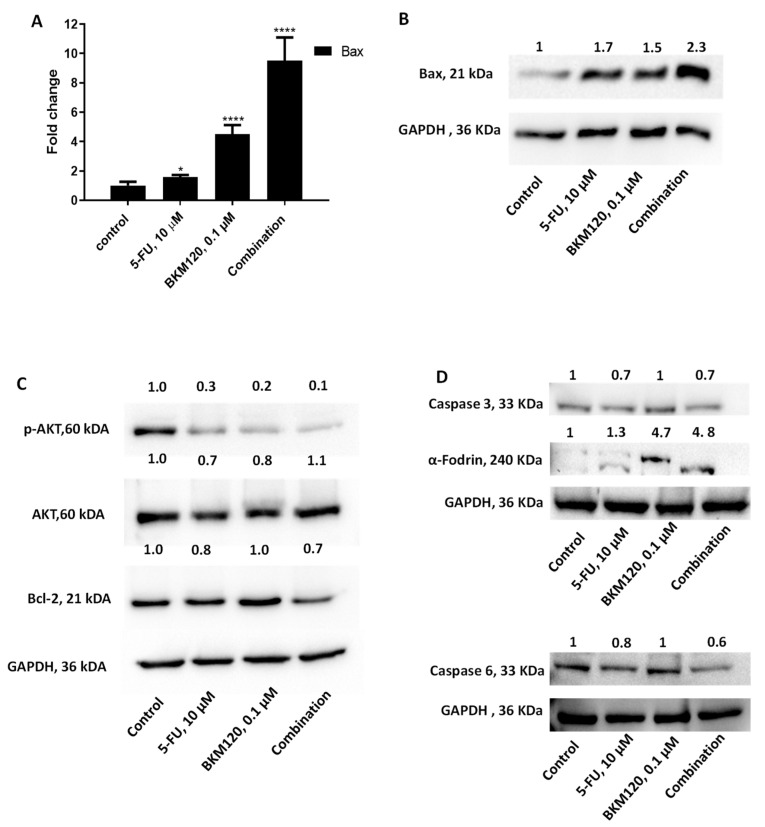
The combination of 5-FU and BKM120 modifies apoptotic protein expression in H460 cells. H460 cells were incubated with 10 μM 5-FU with 0.1 μM BKM120 for 48 h. Combination treatment caused an increase in Bax (**A**) mRNA and (**B**) protein levels. (**C**) Protein expression levels of AKT, p-AKT, and Bcl-2 proteins were measured. (**D**) Full-length caspase-3 and caspase-6 were reduced, while α-Fodrin cleavage was observed. The results are presented as the mean ± SEM resulting from the triplicates of three independent experiments. *p*-value: * <0.05, **** <0.0001.

**Table 1 ijms-26-08378-t001:** IC50 values of Cisplatin, 5-FU, MK2206, and BKM120 in H460 and A549 cell lines. Concentrations are expressed in μM and were calculated by the MTT assay.

**H460**	**24 h**	**48 h**	**72 h**
Cisplatin	50.4	8.6	6.0
5-FU	85.5	105.2	12.3
MK2206	117.5	202.3	N/A
BKM120	2.2	1.2	0.8
**A549**	**24 h**	**48 h**	**72 h**
Cisplatin	109.4	37.3	25.7
5-FU	134.5	110.2	38.6
MK2206	6.2	4.8	13.6
BKM120	13.3	4.8	6.2

**Table 2 ijms-26-08378-t002:** Combination Index (CI) of Cisplatin and 5-FU in combination with BKM120 in H460 cells after 48 h.

**Dose Cisplatin 48 h**	**Dose BKM120 48 h**	**Effect**	**CI**
10.0	0.1	0.56	1.84
10.0	0.5	0.5	1.61
10.0	1.0	0.18	0.22
**Dose 5-FU 48 h**	**Dose BKM120 48 h**	**Effect**	**CI**
10.0	0.1	0.39	0.08
10.0	0.5	0.34	0.16
10.0	1.0	0.35	0.32

**Table 3 ijms-26-08378-t003:** Combination Index (CI) of Cisplatin and 5-FU in combination with BKM120 and MK2206 in A549 cells after 48 h.

**Dose Cisplatin 48 h**	**Dose BKM120 48 h**	**Effect**	**CI**
10.0	1.0	0.54	0.48
**Dose Cisplatin 48 h**	**Dose MK2206 48 h**	**Effect**	**CI**
10.0	1.0	0.64	0.88
**Dose 5-FU 48 h**	**Dose BKM120 48 h**	**Effect**	**CI**
10.0	1.0	0.49	0.23
**Dose 5-FU 48 h**	**Dose MK2206 48 h**	**Effect**	**CI**
10.0	1.0	0.42	0.12

## Data Availability

The original contributions presented in this study are included within this article/Appendix A. Further inquiries can be directed to the corresponding author.

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
