# Peer review of "Enhancing Chemotherapeutic Efficacy in Lung Cancer Cells Through Synergistic Targeting of the PI3K/AKT Pathway with Small Molecule Inhibitors"

_ijms, 2025, doi:10.3390/ijms26178378_

Round 1
Reviewer 1 Report
Comments and Suggestions for Authors
In the present study, the authors investigated the synergistic effects of 5-FU and BKM120, which significantly reduced cell viability and induced apoptosis in NSCLC cells. However, the study is preliminary and requires further research to validate these findings.
Scientific comments—
- BKM120 is already a well-known small molecule inhibitor and is currently undergoing clinical trials. What is the rationale for selecting it for this study?
- The study appears to be quite preliminary. Additional experiments, particularly using an appropriate animal model, are necessary to robustly support your conclusions.
- It would strengthen the study if the Western blot data were supported by quantitative real-time PCR (RT-qPCR) analysis of the targeted genes.
Minor comments:
- Use a uniform pattern for referring to figures throughout the manuscript. Choose either 'Fig.' or 'Figure' and apply it consistently (e.g., 'Fig. 1' or 'Figure 1').
- Use a uniform pattern for referring to hours throughout the manuscript. Choose either 'hrs' or 'h' and apply it consistently (e.g., '24hrs. 1' or '24h').
- Check the reference format of the MS according to the journal and ensure all entries are consistent.
Author Response
Reviewer 1
Scientific comments—
- BKM120 is already a well-known small molecule inhibitor and is currently undergoing clinical trials. What is the rationale for selecting it for this study?
We agree that BKM120 is a well-characterized PI3K inhibitor with documented activity in multiple malignancies and ongoing clinical evaluation. The rationale for selecting it for this study was twofold: First, as a pan-PI3K inhibitor with proven potency and selectivity, BKM120 provides a robust tool to interrogate the therapeutic relevance of PI3K/AKT signaling in NSCLC. Second, both chemotherapeutic drugs that we tested (Cisplatin and 5-FU) act by inducing DNA damage and cancer cells evade their induced cytotoxicity by activating survival signaling pathways, including PI3K/AKT. Blocking PI3K reduces pro-survival signaling, lowers anti-apoptotic defenses (e.g., Bcl-2 family upregulation), and impairs DNA damage repair capacity. Thus, the combination of chemotherapy with a PI3K inhibitor like BKM120 not only enhances cytotoxicity but also overcomes a well-documented mechanism of chemoresistance. We added a clear explanation of the rationale for selecting BKM120 along with appropriate references in lines 399-404 and lines 470-477.
- The study appears to be quite preliminary. Additional experiments, particularly using an appropriate animal model, are necessary to robustly support your conclusions.
We acknowledge the reviewer’s concern and note that our study was designed as an in vitro proof-of-concept to identify synergistic interactions between chemotherapy and PI3K/AKT inhibition in NSCLC. The observed selectivity of the 5-FU and BKM120 combination provides a strong rationale for further work. We have added a statement in lines 598–602 acknowledging that future studies in organoid or in vivo models would be valuable to strengthen the translational relevance of these findings.
3.It would strengthen the study if the Western blot data were supported by quantitative real-time PCR (RT-qPCR) analysis of the targeted genes.
The Western Blot data that appear in Figure 6, include representative images of the proteins Bax (Fig. 6B), which is accompanied by RT-qPCR analysis (Fig. 6A), Bcl-2, p-AKT and corresponding AKT levels (Fig. 6C), full length of proteins Caspases -3 and -6 and full length as well as cleaved levels of α-Fodrin (Fig.6D). We did not assess AKT mRNA levels, as its phosphorylation status is the key determinant of the anti-survival effects under investigation. Similarly, for Caspases and α-Fodrin, cleavage rather than transcript levels serves as the relevant indicator of apoptosis induction. The mRNA levels of Bcl-2 after treatment have now been added as Supplementary Figure 2 and mentioned in the manuscript in lines 346-347.
Minor comments:
- Use a uniform pattern for referring to figures throughout the manuscript. Choose either 'Fig.' or 'Figure' and apply it consistently (e.g., 'Fig. 1' or 'Figure 1').
This has been corrected throughout the manuscript.
2. Use a uniform pattern for referring to hours throughout the manuscript. Choose either 'hrs' or 'h' and apply it consistently (e.g., '24hrs. 1' or '24h').
This has been corrected throughout the manuscript.
3. Check the reference format of the MS according to the journal and ensure all entries are consistent.
All references have been added automatically using the EndNote software according to the IJMS style.
Reviewer 2 Report
Comments and Suggestions for Authors
In this manuscript, the authors combined chemotherapeutic agents (cisplatin and 5-FU) with PI3K/AKT inhibitors (MK2206 and BKM120) to treat non-small cell lung cancer (NSCLC). Using dose–response analysis via MTT assay, they identified a synergistic effect between 5-FU and BKM120, and attributed the in vitro efficacy to apoptosis mediated by Bcl-2 and caspase-3/6 regulation. The study is of potential interest and can be considered for acceptance after the following points are addressed:
- The authors state that “The PI3K/AKT pathway has been identified as a major contributor to 5-FU resistance in lung cancer.” Please expand the discussion to clearly explain the rationale for combining chemotherapies with PI3K/AKT small-molecule inhibitors, and how the 5-FU + BKM120 combination specifically overcomes 5-FU resistance. It is also suggested to assess cytotoxicity in a drug-resistant cell line to strengthen the mechanistic link to resistance.
- Annexin V⁻/PI⁺ cells are generally considered necrotic (primary necrosis or mechanical damage), rather than apoptotic, and are not typically included in apoptosis quantification. If the treatment is intended to induce apoptosis, the focus should be on Annexin V⁺ populations (early and late apoptosis). A high Annexin V⁻/PI⁺ fraction may suggest necrotic cell death or sample handling artifacts. Please review the experimental protocol, repeat the assay if necessary, and ensure that only early and late apoptotic cells are quantified for comparisons.
- It is notable that the 5-FU + BKM120 combination showed no toxicity in MRC5 cells (Fig. 5B). The authors hypothesize that “SMI-induced G0/G1 cell cycle arrest reduces the proportion of cells undergoing DNA synthesis, thereby diminishing the efficacy of S-phase–specific agents such as 5-FU.” Please discuss the cell cycle differences between normal and cancer cells that could explain why normal cells are protected while cancer cells are killed. To support this hypothesis, it is suggested to perform cell cycle analysis on both cell lines following treatment.
- In the ROS assay, cells were incubated with treatments for 30 and 60 min (Fig. 5C), whereas apoptosis analysis was performed after 48 h of incubation. To better interpret the ROS contribution, please use consistent incubation times between assays.
- There are inconsistencies in unit notation (e.g., Page 13, line 473: “20 ul”; Page 14, line 498: “1 μm”). Please check the entire manuscript for correct scientific unit formatting (e.g., μL, μM) and revise accordingly.
Author Response
Reviewer 2
- The authors state that “The PI3K/AKT pathway has been identified as a major contributor to 5-FU resistance in lung cancer.” Please expand the discussion to clearly explain the rationale for combining chemotherapies with PI3K/AKT small-molecule inhibitors, and how the 5-FU + BKM120 combination specifically overcomes 5-FU resistance. It is also suggested to assess cytotoxicity in a drug-resistant cell line to strengthen the mechanistic link to resistance.
We thank the reviewer for this comment. We added a clearer explanation of the rational for combining PI3K/AKT small-molecule inhibitors with chemotherapy in lines 399-404. We also explained the potential underlying mechanism of the effectiveness of the 5-FU + BKM120 combination and how that would specifically overcome 5-FU resistance lines 470-477. The cell lines used as part of our study are not specifically manipulated to be 5-FU resistant; however, both H460 and A549 cells in which the 5-FU + BKM120 combination had a synergistic effect (Table 2, Table 3), are characterized as resistant to 5-FU according to the https://www.cancerrxgene.org/ website. Nevertheless, we agree with the reviewer that a specially manipulated cell line that has developed drug resistance to 5-FU should be explored to better validate the mechanistic effects of drug resistance and its relevance to the PI3K/AKT pathway. We added a comment regarding this in lines 596-598.
2. Annexin V⁻/PI⁺ cells are generally considered necrotic (primary necrosis or mechanical damage), rather than apoptotic, and are not typically included in apoptosis quantification. If the treatment is intended to induce apoptosis, the focus should be on Annexin V⁺ populations (early and late apoptosis). A high Annexin V⁻/PI⁺ fraction may suggest necrotic cell death or sample handling artifacts. Please review the experimental protocol, repeat the assay if necessary, and ensure that only early and late apoptotic cells are quantified for comparisons.
We agree with the Reviewer that only the Annexin V+ cell populations are indicative for apoptosis and have changed the phrasing of the paragraph between lines 289-296 to make this clear. However, since the necrotic population was not significant in the control vs compound treatment in our experiments, we do not believe that detecting this population in our treated samples was a handling artifact. We added a comment on the high presence of the necrotic population in the combination therapy and that it warrants further investigation as the combination therapy may be inducing other types of cell death (lines 296-298).
3. It is notable that the 5-FU + BKM120 combination showed no toxicity in MRC5 cells (Fig. 5B). The authors hypothesize that “SMI-induced G0/G1 cell cycle arrest reduces the proportion of cells undergoing DNA synthesis, thereby diminishing the efficacy of S-phase–specific agents such as 5-FU.” Please discuss the cell cycle differences between normal and cancer cells that could explain why normal cells are protected while cancer cells are killed. To support this hypothesis, it is suggested to perform cell cycle analysis on both cell lines following treatment.
We thank the reviewer for this insightful comment. We added a paragraph on the differences in cell cycle control between MRC-5 cells and cancer cells that could potentially explain the difference in response that we observed following combination treatment. We also referred to the literature to make an informed comparison of the proliferative capacity and the % of cells in the G1 phase between MRC-5 and H460 cells which may potentially affect their response to treatment (lines 457-467).
4. In the ROS assay, cells were incubated with treatments for 30 and 60 min (Fig. 5C), whereas apoptosis analysis was performed after 48 h of incubation. To better interpret the ROS contribution, please use consistent incubation times between assays.
We thank the Reviewer for making this comment. Indeed, the cells were incubated for 48h with 5-FU, BKM120 or their combination, followed by 1h incubation with DCFH-DA and fluorescent measurements were subsequently taken after 10 and 30 minutes. We have corrected this in the Methodology section (lines 518-524) as well as in the description of the Results and Discussion.
5. There are inconsistencies in unit notation (e.g., Page 13, line 473: “20 ul”; Page 14, line 498: “1 μm”). Please check the entire manuscript for correct scientific unit formatting (e.g., μL, μM) and revise accordingly.
This has been corrected throughout the manuscript.
Round 2
Reviewer 2 Report
Comments and Suggestions for Authors
The authors have adequately addressed the concerns. The manuscript is now suitable for publication.